# Developments in Microbial Communities and Interaction Networks in Sludge Treatment Ecosystems During the Transition from Anaerobic to Aerobic Conditions

**DOI:** 10.3390/microorganisms13092178

**Published:** 2025-09-18

**Authors:** Xiaoli Pan, Lijun Luo, Hui Wang, Xinyu Chen, Yongjiang Zhang, Yan Dai, Feng Luo

**Affiliations:** 1Chongqing Key Laboratory of Bioresource for Bioenergy, College of Resources and Environment, Southwest University, Chongqing 400715, China; 18883278242@163.com (X.P.); llj612217@163.com (L.L.);; 2Department of Environment and Quality Test, Chongqing Chemical Industry Vocational College, Chongqing 401220, China

**Keywords:** anaerobic-aerobic transition, microbial succession, co-occurrence network, wastewater treatment

## Abstract

The transition between anaerobic and aerobic conditions represents a fundamental ecological process occurring ubiquitously in both natural ecosystems and engineered wastewater treatment systems. This study investigated the microbial community succession and co-occurrence network dynamics during the transition from anaerobic sludge to aerobic cultivation. High-throughput 16S and 18S rDNA sequencing revealed two distinct succession phases: an initial “aerobic adaptation period” (Day 1) and a subsequent “aerobic stable period” (Day 15). Eukaryotic communities shifted from Cryptomycota to the unassigned eukaryotes dominance, while prokaryotic communities maintained Firmicutes and Proteobacteria as core phyla, with persistent low-abundance archaea indicating functional adaptation. Network analysis highlighted predominant co-occurrence patterns between eukaryotic and prokaryotic communities, suggesting synergistic interactions. These findings provide insights into microbial ecological dynamics during anaerobic-to-aerobic transitions, offering potential applications for optimizing wastewater treatment processes.

## 1. Introduction

Anaerobic digestion, a widely adopted technology for treating swine wastewater with high organic content, offers significant advantages over aerobic biological treatment, particularly in terms of operational efficiency and environmental sustainability, including higher organic loading capacity, reduced sludge yield, lower operational costs, minimized energy consumption, potential for energy recovery, and decreased greenhouse gas emissions [1]. Anaerobic digestion technology demonstrated dual environmental benefits, serving as an efficient organic matter removal system while simultaneously generating renewable energy in the form of biogas, thereby contributing to sustainable waste management practices [2]. However, the anaerobic digestion process generates substantial quantities of by-products, particularly anaerobic digestate, which retains significant concentrations of nutrients (nitrogen and phosphorus), recalcitrant organic carbon compounds, and pathogenic microorganisms, necessitating further treatment and management [3,4]. High-strength wastewater undergoes anaerobic digestion, followed by mixing the digested effluent with low-strength wastewater, and subsequent aerobic treatment through either sequencing batch reactor (SBR) or anoxic/oxic (A/O) processes [5,6,7]. The integration of anaerobic and aerobic processes is industrially essential because the anaerobic phase efficiently converts high-strength organic pollutants into valuable biogas, significantly reducing treatment energy costs and enabling energy recovery, while the subsequent aerobic phase ensures thorough removal of residual organics and nutrients to meet stringent discharge standards [2]. For the treatment of high-strength swine wastewater, a sequential anaerobic–aerobic biological process is typically employed as the most effective treatment strategy [1]. Current research efforts are predominantly directed toward performance optimization and operational parameter refinement of this treatment process, with particular emphasis on enhancing its engineering applicability and practical implementation. For example, in the aerobic treatment phase, raw swine wastewater is strategically introduced to optimize the carbon-to-nitrogen (C/N) ratio, thereby enhancing the biological removal efficiency of both organic and nitrogenous pollutants [5].

However, the transition from anaerobic to aerobic conditions induces significant microbial community restructuring, potentially altering both taxonomic composition and functional dynamics within the ecosystem. The bacterial community composition in anaerobic digestion sludge is predominantly characterized by six major phyla: Firmicutes, Bacteroidota, Proteobacteria, Actinobacteriota, Chloroflexi, and Euryarchaeota, which collectively constitute the core microbial consortium essential for anaerobic degradation processes [8]. Research on eukaryotic communities within anaerobic digestion systems has been relatively limited compared to prokaryotic investigations. Pioneering the investigation of eukaryotic communities in anaerobic sludge through comparative 18S rRNA gene sequence analysis, Matsubayashi et al. provided evidence of Cryptomycota LKM11′s survival, reproduction, and potential functional involvement in anaerobic digestion processes, marking a significant advancement in understanding eukaryotic contributions to anaerobic systems [9]. The aerobic activated sludge system sustains a core functional microbial consortium primarily composed of bacteria, fungi, and protozoa, which collectively facilitate the simultaneous removal of organic carbon, nitrogen, and phosphorus through complex metabolic interactions and ecological relationships. In aerobic sludge systems, prokaryotic microorganisms typically constitute the dominant population, while protozoa serve as crucial components of the microbial food web, fulfilling essential ecological functions and simultaneously acting as sensitive bioindicators of environmental fluctuations [10]. Fungal communities, characterized by their extensive biomass and remarkable diversity, serve as primary decomposers in wastewater treatment systems, playing pivotal roles in organic matter biodegradation and nutrient cycling processes, particularly under specific environmental conditions [11]. The alternation between anaerobic and aerobic conditions represents a fundamental ecological and engineering principle, ubiquitously observed in both natural ecosystems and engineered water treatment systems. Currently, a significant knowledge gap exists regarding the microbial ecological dynamics and underlying mechanisms governing community succession during the transition from anaerobic to aerobic environments. Comprehending the intrinsic microbial community dynamics during the anaerobic-to-aerobic transition is essential for elucidating the functional roles and ecological interactions of diverse microorganisms, thereby advancing our understanding of microbial succession mechanisms in engineered ecosystems. We hypothesize that the introduction of oxygen will be the primary driver of microbial community succession, resulting in a distinct two-phase shift: an initial phase of rapid restructuring during the aerobic adaptation period, followed by a later phase of stabilization into a new, oxygen-adapted community.

This study implements a novel microbial acclimation strategy that initiates with anaerobic sludge inoculation, strategically introduces 1% (*v*/*v*) aerobic sludge inoculum under controlled oxygenation conditions, and employs pig manure anaerobic digestate as substrate to systematically drive microbial community adaptation from anaerobic to aerobic metabolic regimes. By transferring anaerobic sludge to an aerobic environment, the removal abilities of organic matter and nutrients in the anaerobic digestate are enhanced. Through time-series microbial community profiling employing high-throughput 16S/18S rRNA gene amplicon sequencing at critical transition phases, this study systematically elucidates the dynamic succession patterns of microbial diversity and population structure during the anaerobic-to-aerobic transformation, while establishing an intricate interspecies interaction network that governs the functional adaptation processes.

## 2. Materials and Methods

### 2.1. The Source of the Sludge and Anaerobic Stabilization Cultivation

To procure anaerobic sludge and wastewater substrate requisite for the aerobic conversion experiments, three anaerobic reactors were employed, each continuously fed with pig manure wastewater at a chemical oxygen demand (COD) concentration of 15,000 mg/L. The pig manure wastewater was collected from a farm in Beibei District, Chongqing. The hydraulic retention time (HRT) was uniformly maintained at 15 days across all systems, with cyclic feeding and discharging operations conducted at 3-day intervals. Digester operation was consistently carried out under mesophilic conditions at 25 ± 2 °C.

Reactors 1, featuring an effective volume of 10 L, achieved stable operation over a six-month period, following which its anaerobically digested sludge was quantitatively partitioned and transferred into two supplementary reactors, each with an effective volume of 5 L. After 45 days of continuous operation, the matured sludge from Reactor 3 was aseptically transferred to a sequencing batch reactor (SBR) along with 1% aerobic activated sludge for subsequent aerobic enrichment cultivation. The anaerobic and aerobic activated sludge required for the experiment were collected from the anaerobic and aeration tank of Beibei Wastewater Treatment Plant in Chongqing, China, respectively. Meanwhile, the effluent from reactor 2 was appropriately conditioned to achieve standardized concentrations of COD (1500 ± 50 mg/L) and total nitrogen (TN; 300 ± 10 mg/L) for use as synthetic influent during the SBR-based aerobic transition phase (Figure 1).

### 2.2. The Transition Cultivation from Anaerobic Sludge to Aerobic Sludge

The experiment was conducted using Sequencing Batch Reactors (Tevia Acrylic Products Factory, Guangdong, China) under room temperature conditions (25 ± 2 °C), as shown in Figure 1. Each SBR had a diameter of 16 cm, effective height of 25 cm, and working volume of 5 L. The influent consisted of anaerobically digested effluent of reactor 2 characterized by a COD of 1500 ± 50 mg/L and total nitrogen (TN) concentration of 300 ± 10 mg/L. The influent was introduced through the bottom inlet of the SBR, while the treated effluent was discharged from the upper outlet at a 50% volume exchange rate. The anaerobic sludge collected from the anaerobic digestion reactor treating pig manure wastewater was used as the seed sludge, with the seeding sludge occupying 20% of the working volume. The reactors operated in batch mode with a cycle of 12 h, which consisted of influent (10 min), aeration (660 min), settling (30 min), effluent (10 min), and idle (10 min) phases. The reactor is aerated using an air pump, and the dissolved oxygen concentration is controlled using a gas flow meter while being monitored with a dissolved oxygen meter to maintain it at 2–3 mg/L.

### 2.3. Measurement and Analysis

#### 2.3.1. Components Analysis

During the reactor operation period, the influent and effluent samples were taken and analyzed every day. Before conducting the analysis, each influent and effluent sample was filtered through a 0.45 μm polyethersulfone membrane. Ammonia nitrogen (NH_4_^+^-N), nitrate nitrogen (NO_3_^−^-N), nitrite nitrogen (NO_2_^−^-N) and COD were measured according to standard methods [12]. Additionally, the temperature and pH of the influent and reactors were regularly measured. The tests were conducted in triplicate, with results presented as the mean values.

#### 2.3.2. DNA Extraction and 16S rDNA,18S rDNA Sequencing Analysis

Based on the response of COD and NH_4_^+^-N removal rates, DNA extraction from sludge was performed at six different time points, labeled as D0 (anaerobic phase), D1, D3, D5 (early aerobic phase), D15, and D30 (stable aerobic phase). Total DNA was extracted from 2 mL of sludge using a Power Soil DNA Isolation Kit (MO BIO Laboratories, Carlsbad, CA, USA). The quality and quantity of the extracted DNA were determined by 1% agarose gel electrophoresis (Bio-Rad, Hercules, CA, USA) and a NanoDrop-2000 Spectrophotometer (NanoDrop Technologies, Wilmington, DE, USA). The 16S rDNA sequencing and 18S rDNA sequencing were performed using primers 515F/806R and V4_1f/TAReukREV3R, respectively, to amplify the V4 region [13,14]. The sequencing was conducted using Illumina MiSeq sequencing technology (Illumina, San Diego, CA, USA). The sequences obtained from the sequencing were analyzed at the level of individual nucleotide differences using Amplicon Sequence Variants (ASV). The microbial community structure and relative abundance were obtained by ASVs with an online data processing platform developed by Shanghai Majorbio (https://cloud.majorbio.com/, accessed on 13 November 2023).

#### 2.3.3. Neutral Community Model

To evaluate the influence of stochastic processes on microeukaryotic plankton community assembly, we employed a neutral community model, utilizing nonlinear least-squares regression to optimize the fit between ASV occurrence frequency and relative abundance. R^2^ value indicates the goodness of fit to the model, which was calculated following the “Östman’s method” [15]. The nonlinear least-squares regression method was employed to establish the optimal fit between ASV occurrence frequency and relative abundance. The goodness-of-fit (R^2^) was calculated using the “MicEco” package. An R^2^ value approaching 1 indicates that the community assembly is predominantly governed by stochastic processes. Conversely, when the model fails to adequately describe the community composition, R^2^ values may approach or fall below 0. Model computations were performed with R version 3.6.1.

#### 2.3.4. Co-Occurrence Network Analysis

Based on the ASV abundance information obtained from 16S rDNA and 18S rDNA sequencing, only taxa with a total relative abundance greater than 0.5% were retained. The Spearman method was used to calculate the correlation between species, and the *p*-value correction for correlation was performed using the BH method. The topological properties of the co-occurrence network were calculated using the iGraph package in R software, and network visualization was accomplished using Gephi 0.10.1.

## 3. Results

### 3.1. The Removal Performance of Pollutants During the Transition from Anaerobic Sludge to Aerobic Cultivation in the SBR

When anaerobic sludge was inoculated in the SBR and subjected to aerobic transition cultivation, the overall COD removal efficiency was relatively high (approximately 90%) after an acclimation period of about 5 days (Figure 2a), which is consistent with the findings reported by Sun et al. [16]. These findings indicated that an aerobic environment was likely established by the fifth day, potentially triggering corresponding adaptations in the microbial community structure. During the initial 30-day period, nitrification activity was virtually undetectable (Figure 2b), which is consistent with the limited presence of aerobic ammonia-oxidizing bacteria in the anaerobic sludge [17]. The removal efficiency of NH_4_^+^-N was very low on the first day (12%). As the sludge gradually adapted, the removal efficiency reached around 40% by the fifth day and remained at this level until approximately the 30th day. Subsequently, the removal efficiency demonstrated a consistent upward trend, suggesting that microbial community stabilization was achieved by approximately day 30. Ultimately, the removal efficiency of NH_4_^+^-N could reach approximately 95% (Figure 2c). Accordingly, nitrate accumulation was observed in the effluent.

### 3.2. Microbial Community Succession During Aerobic Adaptation of Anaerobic Sludge

#### 3.2.1. Microbial Community Diversity

The efficient removal of both COD and nitrogen indicated successful establishment of an aerobic microbial community within the reactor. The Sobs index in Alpha diversity reflects community richness, which represents the actual number of observed species. The Shannon index reflects community diversity, where a higher Shannon value indicates greater community diversity. Figure 3a revealed that both eukaryotic and prokaryotic communities exhibited increasing trends in Sobs and Shannon indices, which were subsequently followed by gradual declines. Additionally, the Sobs index and Shannon index of prokaryotes are higher than those of eukaryotes. Relative to D0, both microbial richness and diversity increased during the initial phase (D1–D5), potentially attributable to enhanced oxygen utilization for energy generation under aerobic conditions. This enhancement stimulates microbial proliferation and reproduction, consequently augmenting community richness and diversity. However, these parameters declined at D15 before reaching stabilization by D30.

The similarity and dissimilarity of community compositions between different samples were analyzed using Principal Co-ordinates Analysis (PCoA). As illustrated in Figure 3b and Figure 3c, the results demonstrate that the contributions of PC1 and PC2 for eukaryotes are 47.18% and 33.73%, respectively, while the corresponding values for prokaryotes are 52.17% and 23.37%. Compared to D0, there are significant differences in the community structure of D1. The distances between D1, D3, and D5 are close, indicating a clear similarity in community structure. The community structure of D15 undergoes another change, showing significant differences compared to D5. However, D15 and D30 cluster together, suggesting that they have similar community structures. Consistent with our findings, Sun et al. reported bacterial community profiles during the initial phase of aerobic adaptation in anaerobic granular sludge systems, with microbial divergence becoming evident after 18 days of operation [16]. These findings indicated that microbial community structure undergoes two distinct successional transitions during the aerobic adaptation process, with significant restructuring events occurring at day 1 (D1) and day 15 (D15).

#### 3.2.2. Eukaryotic Microbial Community Based on 18S rRNA Gene Sequencing

Figure 4a illustrated the eukaryotic community composition at the phylum level, revealing significant structural shifts during the aerobic adaptation process. In the initial anaerobic sludge sample (D0), Cryptomycota represented the dominant phylum, accounting for 76.85% of the relative abundance. During the aerobic process (D1–D30), Cryptomycota exhibited a substantial decline in relative abundance, ultimately reaching 13.51% by day 30 (D30). This finding aligns with previous studies demonstrating Cryptomycota’s prevalence in anaerobic and hypoxic environments, including activated sludge systems for wastewater treatment, anaerobic digesters, acid mine drainage, and anoxic sediments [9,18,19]. By day 30 (D30), some unassigned (unclassified or no ranked) eukaryotic microorganisms became the dominant phylum, with their relative abundance increasing significantly from 5.10% (D0) to 56.65% (D30). Comprehensive investigations are required to elucidate the phylogenetic relationships, taxonomic classification, and functional characteristics of these unassigned eukaryotic taxa. Nematozoa exhibited relative abundance ranges of 2.31–9.45%. and commonly observed in wastewater treatment systems, demonstrate significant metabolic versatility and growth potential [20,21].

At the genus level (Figure 4b), *Cryptosporidium* and *Colpoda* were the top two most abundant genera. Notably, a significant proportion of eukaryotic microorganisms remained unassigned (unclassified or no ranked), accounting for 66.64% to 84.60% of the total, with the highest proportion (84.60%) observed in D0. This is primarily due to the fact that the vast majority of eukaryotic microorganisms cannot be cultured under laboratory conditions, and their taxonomic research remains relatively underdeveloped. As a result, while their presence can be detected through DNA sequencing, formal classification remains challenging. These findings also suggest the presence of a large number of novel, functionally unknown microorganisms in the sludge treatment ecosystem. Previous studies have demonstrated that *Cryptomycota* maintains a significantly higher relative abundance in anaerobic digested sludge systems compared to other microbial communities [10]. *Cryptosporidium* is frequently identified in wastewater treatment systems, where aeration processes appear to facilitate its persistent colonization within sludge communities [22]. Furthermore, *Cryptosporidium* demonstrated the capacity to form independent biofilms or establish symbiotic associations with bacterial biofilms, particularly those produced by *Pseudomonas aeruginosa*, thereby enhancing its environmental persistence [23,24]. Previous studies have successfully isolated Colpoda species from activated sludge systems in wastewater treatment plants [25]. Under typical environmental conditions, *Colpoda* species primarily exist as free-swimming organisms with limited biofilm colonization capacity, resulting in their sporadic occurrence in microbial communities [26].

At the order level (Figure A1), certain unassigned eukaryotic microorganisms (norank_p__Cryptomycota and unclassified_d__Eukaryota) have always accounted for a large proportion in terms of abundance. In the aerobic condition D15, compared to the anaerobic condition D0, the relative abundance of the eukaryote *norank_p__Cryptomycota* (Figure A2a) significantly decreased. In contrast, the relative abundance of *unclassified_d__Eukaryota* and *Colpoda* significantly increased. Further research and analysis are needed to characterize these unassigned eukaryotes.

#### 3.2.3. Prokaryotic Microbial Community Based on 16S rRNA Gene Sequencing

Figure 5a illustrated the prokaryotic community composition at the phylum level, revealing Firmicutes and Proteobacteria as the dominant phyla throughout the aerobic adaptation process, collectively representing approximately 80% of the total sequenced reads. The relative abundance of Firmicutes exhibited an initial decline from 87.60% (D0) to 49.97% (D3), followed by a subsequent recovery to 72.26% by day 30 (D30). Members of the Firmicutes phylum are predominantly aerobic or facultative anaerobic bacteria capable of producing various enzymes that degrade organic matter in wastewater [27]. It is therefore reasonable to observe their higher abundance in anaerobic dissolved oxygen (DO) environments. In contrast, Proteobacteria demonstrated a consistent upward trend in relative abundance, increasing from 1.11% to 11.85% throughout the experimental period. Certain members of the Proteobacteria phylum possess the ability to degrade cellulose and proteins, and their overall relative abundance is generally higher in aerobic environments than in strictly anaerobic conditions, primarily due to the efficient oxygen-dependent metabolism of their numerous aerobic or facultative anaerobic constituents [28]. The phyla Bacteroidota, Actinobacteriota, and Chloroflexi exhibited relative abundance ranges of 2.09–5.22%, 1.71–5.30%, and 0.77–3.43%, respectively, throughout the experimental period. Bacteroidota, Actinobacteriota, and Chloroflexi usually collectively constituted the core microbial consortium essential for anaerobic degradation processes [8]. Within the archaeal domain, Halobacterota exhibited a relative abundance range of 0.80–6.58%. This finding aligns with previous studies reporting Halobacterota detection in aerobic sludge systems [29]. Notably, trace amounts of Euryarchaeota were consistently detected throughout the process. Previous studies have identified Euryarchaeota as the predominant archaeal phylum in flocculent sludge systems [30]. Remarkably, Euryarchaeota demonstrate oxygen tolerance, maintaining viability in sludge systems despite aerobic conditions. This observation is supported by previous studies confirming archaeal persistence in aerobic sludge environments [31].

The bacterial community was dominated by five genera: *Clostridium_sensu_stricto_1* (21.52–37.05%), *Acinetobacter* (0.92–23.16%), *Terrisporobacter* (8.31–12.90%), *Romboutsia* (4.90–8.78%), and *Turicibacter* (3.26–8.43%), as shown in Figure 5b. In a study investigating nitrite nitrogen removal from piggery wastewater using a sequencing batch reactor (SBR), Qi et al. identified *Clostridium* as one of the dominant bacterial genera under controlled conditions of 25 °C and 2 mg/L dissolved oxygen throughout the treatment process [32]. *Acinetobacter* exhibited comparative enrichment by day 1 (D1), achieving a peak relative abundance of 21.17%, followed by a substantial decline to 0.92% by day 30 (D30), with the most pronounced decrease occurring by day 15 (D15). *Acinetobacter* are recognized for their dual metabolic capabilities in heterotrophic nitrification and aerobic denitrification processes [33]. As a facultative anaerobic genus, *Terrisporobacter* exhibits remarkable oxygen tolerance and adaptability to varying dissolved oxygen concentrations. This genus plays a crucial role in organic matter decomposition and potentially establishes syntrophic relationships with methanogenic archaea [34]. *Romboutsia*, a hydrogen-producing bacterium, facilitates pollutant degradation through hydrogen-mediated electron transfer, thereby enhancing the overall degradation efficiency [35]. *Turicibacter* potentially plays a significant role in organic matter decomposition and nitrogen transformation processes, with particular involvement in ammoniacal nitrogen conversion, thereby contributing to nitrogen removal and cycling efficiency [36]. Comparative analysis between aerobic (D15) and anaerobic (D0) conditions revealed significant shifts in bacterial community composition (Figure A2b). The relative abundance of several taxa showed marked reduction under aerobic conditions, including *Turicibacter*, *Romboutsia*, *Tissierella*, *Terrisporobacter*, *Clostridium_sensu_stricto_1*, and *Proteiniphilum*. Conversely, significant increases were observed in *Acinetobacter*, *norank_f__Syntrophomonadaceae*, *unclassified_f__Rhodobacteraceae*, and *Fastidiosipila*. Concurrently, methanogenic archaea, predominantly comprising *Methanosarcina*, *Methanobacterium*, and *Methanococcus*, were detected at a relative abundance of approximately 5%. Liu et al. identified methanogenic archaea in sludge samples from a large-scale SBR wastewater treatment plant [30]. Remarkably, these archaea demonstrate long-term viability despite prolonged exposure to aerobic conditions.

### 3.3. Process of Microbial Community Formation

In this study, a neutral community model was used to predict the relationship between the occurrence frequency of ASVs and their relative abundance in the eukaryotic and prokaryotic microbial communities during the transition of anaerobic sludge to aerobic environments (Figure 6). In the figure, R^2^ represents the goodness of fit of the neutral community model. A higher R^2^ value indicates that the community assembly process is more influenced by randomness, while a lower R^2^ value indicates a greater influence of determinism. Stochastic processes are governed by ecological equivalence among species and random events including birth, death, dispersal, extinction, and speciation. In contrast, deterministic processes are driven by biotic interactions (e.g., competition, predation, and symbiosis) and abiotic environmental factors (e.g., temperature, pH, and salinity). These factors collectively govern the assembly and succession of microbial communities through complex interactions [37]. Consequently, stochastic processes can be conceptualized as probabilistic equilibria between microbial immigration and extinction events, whereas deterministic processes represent systematic microbial adaptations to environmental constraints.

Figure 6 demonstrated that the neutral community model provides a statistical estimation of the frequency–abundance relationship for ASVs within the microbial community. Throughout the aerobic adaptation of anaerobic sludge, the influence of stochastic processes progressively diminished, as evidenced by decreasing R^2^ values from 0.698 to 0.381 in eukaryotic communities and from 0.814 to 0.737 in prokaryotic communities, corresponding to the transition from initial aerobic adaptation to stable operational phases. These findings demonstrate an increasing predominance of deterministic processes in governing the assembly of both eukaryotic and prokaryotic microbial communities [38]. These results suggest progressive adaptation of both eukaryotic and prokaryotic communities to aerobic conditions throughout the experimental period.

Canonical correspondence analysis was performed to examine the relationships between microbial communities (eukaryotic and prokaryotic) and physicochemical properties, as shown in Figure A3a and Figure A3b, respectively. The first two RDA axes (RDA1 and RDA2) collectively explained 81.65% and 79.77% of the total variation in eukaryotic and prokaryotic microbial communities, respectively. These results demonstrate that COD and NH_4_^+^-N, served as the primary environmental drivers shaping the community structures of both eukaryotic and prokaryotic microorganisms.

### 3.4. Analysis of Microbial Co-Occurrence Networks

Microorganisms establish complex ecological networks through competitive, cooperative, and symbiotic interactions across diverse habitats. Utilizing ASV abundance data derived from 16S and 18S rDNA sequencing, we constructed microbial co-occurrence networks, supported by PCoA analysis results (Figure 4). To investigate microbial interaction dynamics during aerobic adaptation, we analyzed samples representing two distinct phases: the initial aerobic phase (D0, D1, D3, D5) and the stabilized aerobic phase (D15, D30), capturing the transition from anaerobic to aerobic conditions in sludge communities. As shown in Table A1, the eukaryotic co-occurrence network during the initial aerobic phase comprises 144 nodes and 543 edges, whereas the network in the stabilized aerobic phase expands to 155 nodes and 678 edges. For prokaryotic communities, the co-occurrence network during the initial aerobic phase comprises 434 nodes and 2467 edges, which expands to 452 nodes and 3472 edges in the stabilized aerobic phase. Notably, the average degree increased from 7.54to 8.75 for eukaryotic communities and from 11.37 to 15.36 for prokaryotic communities during the transition from initial to stabilized aerobic phases. These findings demonstrate that microbial networks in the stabilized aerobic phase exhibit enhanced topological complexity and structural stability [39]. Both eukaryotic and prokaryotic networks exhibited a predominance of positive over negative edges, suggesting that cooperative interactions outweigh competitive exclusion during aerobic adaptation of sludge microbial communities [40]. This phenomenon may be attributed to the co-occurrence of microorganisms with similar ecological niches and environmental preferences, resulting in positive interactions within the meta-community [41]. The prokaryotic network exhibited higher average degree and shorter average path distance compared to its eukaryotic counterpart, suggesting stronger interspecies connectivity and more rapid environmental responsiveness within prokaryotic communities [41].

Microbial co-occurrence networks for the different phases were visually colored according to the phylum of the network nodes (Figure 7). The size of each node was proportional to the number of connections; the edge thickness of each connection between two nodes was proportional to the correlation coefficient. Within eukaryotic communities, unassigned eukaryotic microorganisms (*norank_p__Cryptomycota* and *unclassified_d__Eukaryota*) and Cryptomycota emerged as predominant taxa. Within prokaryotic communities, Firmicutes and Proteobacteria emerged as predominant phyla, a pattern corroborated by relative abundance profiling in microbial community analyses. During the initial aerobic adaptation phase, negative correlations between eukaryotes account for 20.44%, with Cryptomycota and unclassified SAR_k__norank lineages emerging as predominant contributors to these negative associations. Within prokaryotic communities, 35.08% of observed correlations exhibited negative relationships, primarily associated with Firmicutes and Bacteroidota. Notably, these microbial interactions appear to transcend strict interspecific boundaries, instead being mediated by metabolic interdependencies among functional guilds within the ecological network [42].

These findings demonstrate that during the aerobic adaptation of sludge systems, microbial co-occurrence networks exhibited significant increases in mutualistic interactions (positive correlations), topological complexity (node density and edge connectivity), and average node connectivity degree. Cross-domain symbiotic associations between eukaryotic and prokaryotic consortia were significantly enhanced during this transition phase, driving substantial improvements in the network’s topological complexity and dynamic stability.

## 4. Discussion

The transition from anaerobic to aerobic conditions induced significant restructuring of microbial communities, aligning with our hypothesis that oxygen availability would drive distinct successional phases. The observed two-phase succession—characterized by an initial aerobic adaptation period (day 1) followed by stabilization (day 15)—mirrors findings from Sun et al. [16], who reported similar microbial divergence in granular sludge systems. The rapid decline of Cryptomycota and the concurrent rise of unclassified_d__Eukaryota in eukaryotic communities (Figure 4 and Figure A1) indicate that aerobic conditions selectively promote taxa possessing oxidative metabolic capabilities, aligning with their functional roles in organic matter decomposition under oxygenated environments [11]. In contrast, Cryptomycota predominantly thrives in anaerobic and hypoxic habitats—such as activated sludge systems, anaerobic digesters, anoxic sediments, and acid mine drainage—where it frequently constitutes a major eukaryotic component [9,18,19].The persistence of Firmicutes and Proteobacteria throughout the transition, alongside detectable archaeal populations (Halobacterota, Euryarchaeota), supports the hypothesis that certain anaerobic taxa exhibit functional adaptability to aerobic niches, possibly through dormancy or metabolic versatility [30,31]. Specifically, the relative abundance of Firmicutes showed a trend of “initial increase followed by decrease” during the succession process, indicating its competitive advantage during the transition phase. In contrast, Proteobacteria demonstrated a “first decrease then increase” resilience pattern, further highlighting the differential adaptive strategies employed by these taxa under shifting redox conditions. Notably, archaea—particularly Euryarchaeota—exhibit comparable adaptive strategies. This group thrives not only in saline–alkaline conditions but also shows increased diversity in oil-polluted soils, outperforming even pristine environments [43,44]. These findings suggest that Euryarchaeota possess substantial metabolic versatility, enabling persistence under diverse stresses—including oxygen shifts, salinity, and hydrocarbons—thereby challenging the view of archaea as strictly anaerobic specialists.

The shift from stochastic to deterministic processes in community assembly, evidenced by declining neutral model R^2^ values, underscores the increasing influence of environmental filtering and biotic interactions as aerobic conditions stabilize. This aligns with Zhou and Ning [37], who highlighted deterministic dominance in structured environments. The co-occurrence network analysis revealed enhanced mutualistic interactions and topological complexity in stabilized aerobic phases, suggesting that microbial synergies (e.g., cross-domain partnerships between Proteobacteria and eukaryotic decomposers) underpin functional resilience, as proposed by Chen et al. [40]. The prevalence of positive correlations (more than 90% in stable phase) contrasts with the competitive dynamics observed in anaerobic systems, implying that aerobic niches promote cooperative nutrient cycling.

These findings advance the understanding of microbial ecology in engineered systems, highlighting the importance of transition-phase management in wastewater treatment to optimize pollutant removal. The persistence of archaea and facultative anaerobes (*Terrisporobacter*, *Acinetobacter*) suggests their potential as bioindicators for process stability. Future research should (1) elucidate metabolic pathways via metatranscriptomics to resolve the functional roles of unassigned taxa (e.g., unclassified_d__Eukaryota). (2) Explore bioaugmentation strategies using taxa with dual anaerobic–aerobic adaptability (e.g., Romboutsia) to enhance system resilience. (3) Investigate scaling effects by validating these dynamics in full-scale reactors with variable organic loads.

This study bridged ecological theory and engineering practice, offering a framework to tailor microbial consortia for efficient anaerobic–aerobic transitions in wastewater treatment and beyond.

## 5. Conclusions

This study elucidated the microbial community succession and co-occurrence network dynamics during the transition from anaerobic sludge to aerobic cultivation. The findings revealed two distinct succession phases: an initial “aerobic adaptation period” (day 1) and a subsequent “aerobic stable period” (day 15). Eukaryotic communities exhibited a marked shift in dominance from Cryptomycota to unclassified_d__Eukaryota, while prokaryotic communities maintained Firmicutes and Proteobacteria as core phyla, with persistent low-abundance archaea suggesting functional adaptation to oxygenated conditions. The COD and NH_4_^+^-N served as the primary environmental drivers shaping the community structures of both eukaryotic and prokaryotic microorganisms. Network analysis highlighted predominant co-occurrence patterns between eukaryotic and prokaryotic communities, indicating synergistic interactions that enhanced system stability. The results underscore the resilience and adaptability of microbial consortia during environmental transitions, with deterministic processes increasingly governing community assembly as aerobic conditions stabilized. The study provided critical insights into microbial ecological dynamics, offering a foundation for optimizing wastewater treatment processes. Future research should focus on elucidating the functional roles of unclassified eukaryotic taxa and the mechanisms underlying archaeal survival in aerobic environments. These advancements could further refine engineered systems for efficient pollutant removal and sustainable wastewater management.

## Figures and Tables

**Figure 1 microorganisms-13-02178-f001:**
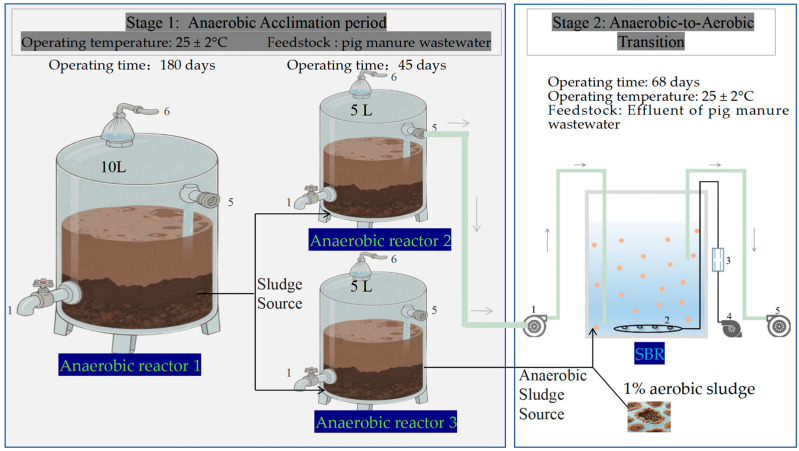
Schematic diagram of the anaerobic acclimation phase and the aerobic transition phase (1: influent pump; 2: micro-porous aeration diffuser; 3: gas flow meter; 4: air pump; 5: effluent pump; 6: biogas collection).

**Figure 2 microorganisms-13-02178-f002:**
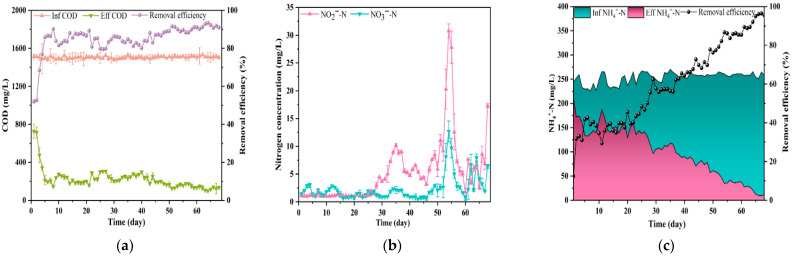
Physicochemical performance during the transition from anaerobic to aerobic process. (**a**) Influent and effluent COD concentrations and COD removal efficiency; (**b**) concentrations of NO_3_^−^-N and NO_2_^−^-N in the effluent; (**c**) influent and effluent NH_4_^+^-N concentrations and NH_4_^+^-N removal efficiency.

**Figure 3 microorganisms-13-02178-f003:**
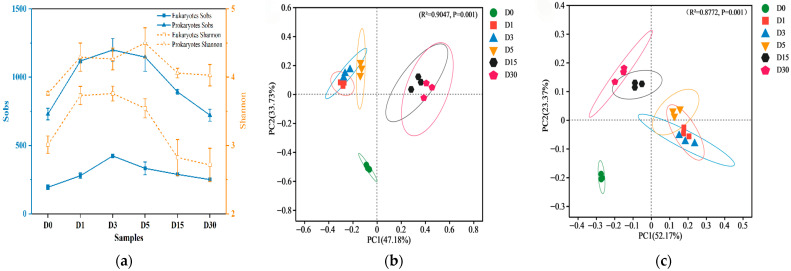
Alpha diversity and PCoA analysis based on ASV classification level. (**a**) Alpha diversity analysis; (**b**) PCoA analysis of eukaryotes; (**c**) PCoA analysis of prokaryotes.

**Figure 4 microorganisms-13-02178-f004:**
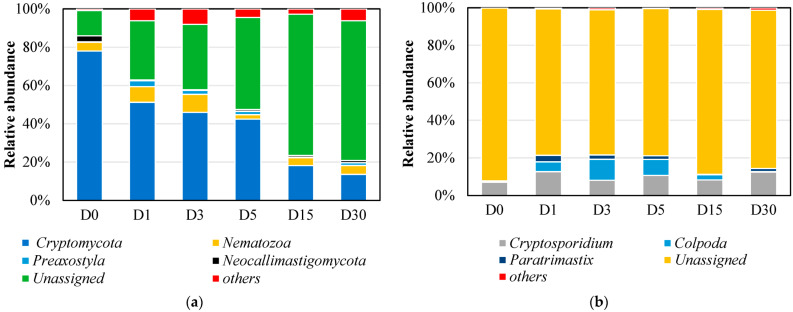
Analysis of eukaryotic microbial community structure composition. (**a**) Phylum level; (**b**) genus level.

**Figure 5 microorganisms-13-02178-f005:**
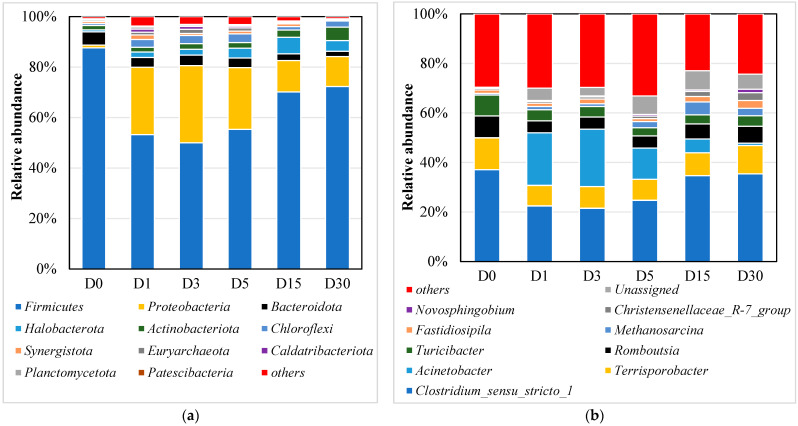
Analysis of prokaryotic microbial community structure composition. (**a**) Phylum level; (**b**) genus level.

**Figure 6 microorganisms-13-02178-f006:**
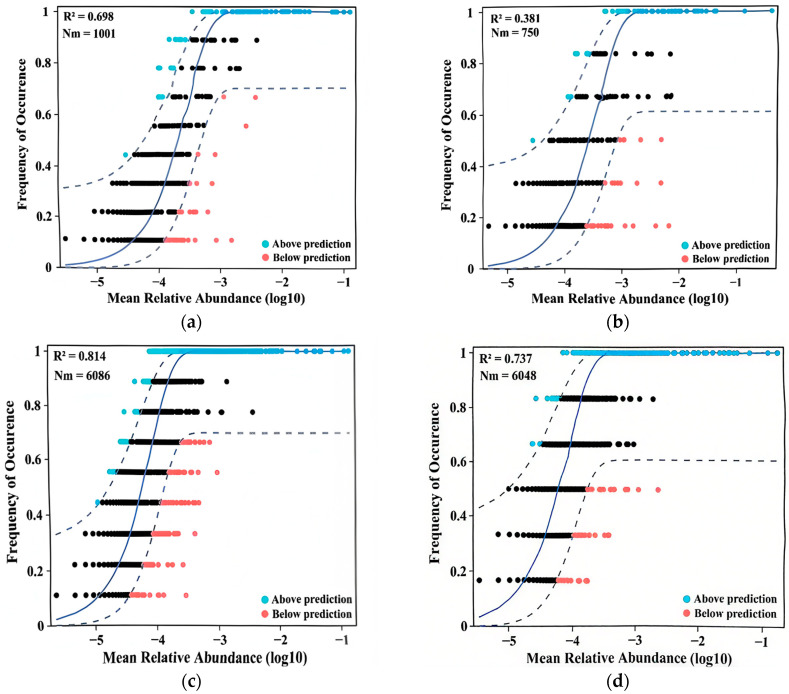
Neutral community model of aerobic initial phase (**a**,**c**) and aerobic stable phase (**b**,**d**). R^2^ represents the fit of the neutral community model, Nm represents the product of the abundance of all ASVs in each sample with the migration rate (m), the solid blue line represents the best fit curve, the dashed blue lines represent the 95% confidence interval, the green and red circles represent ASVs that occurred more and less frequently than predicted. (**a**,**b**) Eukaryotes; (**c**,**d**) prokaryotes.

**Figure 7 microorganisms-13-02178-f007:**
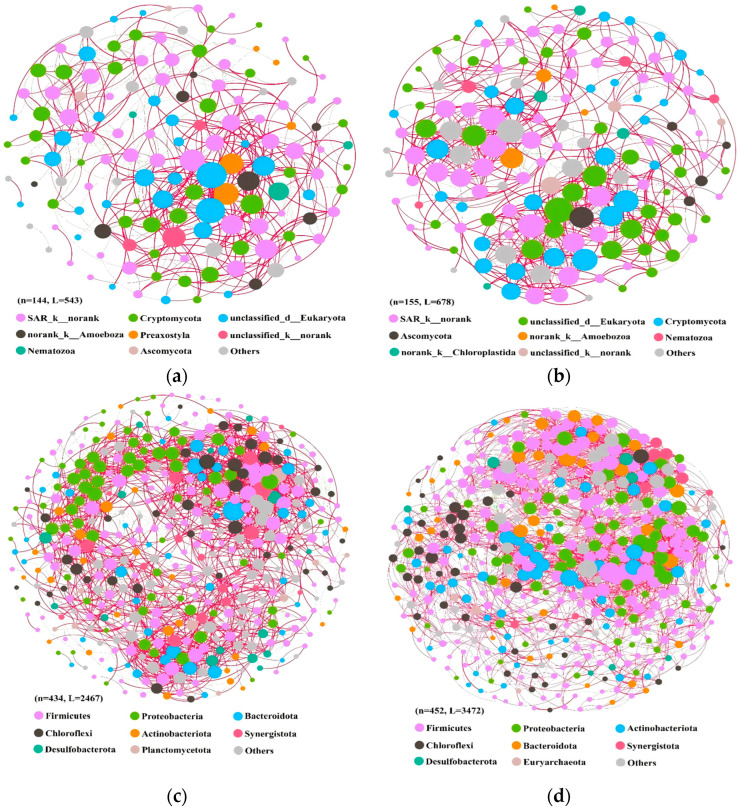
Aerobic initial phase (**a**,**c**) and aerobic stable phase (**b**,**d**) ASV co-occurrence networks. The color of the nodes represents the phylum to which the ASVs belong, and the size of the nodes indicates the number of edges they are connected to (i.e., node degree). The color of the lines represents positive correlation (red) and negative correlation (green), respectively. (**a**,**b**) Eukaryotes; (**c**,**d**) prokaryotes.

## Data Availability

The original contributions presented in this study are included in the article. Further inquiries can be directed to the corresponding author. The 16S rDNA and 18S rDNA sequencing datasets have been submitted to the National Center for Biotechnology Information (NCBI; www.ncbi.nlm.nih.gov) with SRA: PRJNA1287089.

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
