# Peer review of "Developments in Microbial Communities and Interaction Networks in Sludge Treatment Ecosystems During the Transition from Anaerobic to Aerobic Conditions"

_microorganisms, 2025, doi:10.3390/microorganisms13092178_

Round 1
Reviewer 1 Report
Comments and Suggestions for Authors
General Comments:
This manuscript explores an interesting topic that could be of significant interest to the field. However, there are several fundamental aspects that require substantial revision before it can be considered suitable for publication. My primary concerns revolve around the experimental design and the analysis and presentation of taxonomic data.
First, the study's central claim of evaluating the effects of an anaerobic-aerobic transition on microbial communities appears to be undermined by the experimental setup. Based on the description, the experiment was initiated and maintained under aerobic conditions from Day 0, utilizing only an anaerobic inoculum. To genuinely assess the impact of a transition, the experimental design should ideally involve monitoring microbial communities under established anaerobic conditions before introducing aerobic conditions. As it stands, the current setup seems to primarily evaluate how an anaerobic inoculum adapts to and develops within an aerobic environment, rather than a true anaerobic-to-aerobic transition. This distinction is critical and should be accurately reflected in the study's stated objectives, methodology, and interpretation of results. It significantly impacts the interpretation of "microbial succession" in this context. Second, the taxonomic data analysis and presentation require significant re-evaluation. The inclusion of "unclassified" and "no rank" information in figures and discussions, particularly as if they represent distinct taxa, is problematic and diminishes the clarity and interpretability of the results. As a best practice, such unclassified data should be appropriately grouped or excluded from taxonomic analyses when a higher resolution is desired.
Given the extent of these concerns, particularly regarding the experimental design and data presentation, I recommend rejection of the manuscript in its current form. However, I strongly encourage the authors to undertake a major and critical restructuring of the work, addressing these comments comprehensively, and to resubmit it for consideration in the future.
Major comments:
To address these concerns and improve the manuscript's overall quality, I suggest the following:
- Refine Experimental Design and Interpretation:
- Re-evaluate whether the current experimental design truly represents an "anaerobic-aerobic transition." If not, consider rephrasing the study's objective and discussion to accurately reflect that it investigates the adaptation of an anaerobic inoculum to aerobic conditions.
- If the intention is indeed to study a true transition, a revised experimental setup that includes an initial anaerobic phase for community establishment would be necessary.
- Improve Taxonomic Data Analysis and Presentation:
- Figures 4a and 4b: These figures should be re-analyzed and presented to show only taxa with clear taxonomic resolution (e.g., at the Phylum level). All "unclassified," "no rank," or similarly uninformative classifications should be grouped together or excluded. For instance, "Cryptomycota" is a clade, not a phylum, and should be corrected if presented as such.
- Figures 5, 6, and 7: These figures also suffer from low resolution, making them difficult to read. Please ensure high-resolution images are provided.
- Figure 5 Legend: Clean up the legend by removing "unclassified," "no rank," or any other non-specific information, focusing only on classified genera.
- Data Re-analysis: I strongly recommend a thorough re-analysis of the taxonomic data to improve its presentation, focusing on ASVs that demonstrate clear taxonomic resolution at relevant levels (e.g., phylum and genus). The results section should then be rewritten to reflect these improved analyses.
Minor comments:
- Introduction Question (Lines 84-86): The question "What are the ecological dynamics of diverse microbial communities during the transition from anaerobic to aerobic conditions, and how do these transformations influence microbial fate?" is currently too broad. If it represents the core research question, it should be clearly stated as such and directly linked to the specific aims of this study. Otherwise, if it's a general inquiry, it might be better omitted.
- Östman's Method (Line 164): Please add a specific reference for the "Östman's method" cited in the methodology section.
- Hypothesis (Line 424): The study's hypothesis should be clearly stated in the introduction section to set the context for the research.
- Discussion: The discussion section will undoubtedly benefit from significant improvement once the taxonomic information is accurately re-analyzed and presented. It should then critically interpret the findings in light of the refined experimental context and taxonomic data.
Author Response
Comments and suggestions from the reviewer 1:
Major comments:
To address these concerns and improve the manuscript's overall quality, I suggest the following:
Refine Experimental Design and Interpretation:
- Re-evaluate whether the current experimental design truly represents an "anaerobic-aerobic transition." If not, consider rephrasing the study's objective and discussion to accurately reflect that it investigates the adaptation of an anaerobic inoculum to aerobic conditions. Ifthe intention is indeed to study a true transition, a revised experimental setup that includes an initial anaerobic phase for community establishment would be necessary.
Response: Thank you once again for your valuable feedback. We did conduct an initial anaerobic acclimation process for the sludge, which involved three anaerobic fermentation reactors and lasted for over seven months. Subsequently, the anaerobic sludge was transferred to an aerobic environment to investigate its transformation under aerobic conditions (lines 122-125, page 4). To more clearly illustrate the experimental process, we have redesigned Figure 1 (line 149, page 5), which now includes a detailed depiction of both the anaerobic acclimation and aerobic transformation stages.
- (1) Improve Taxonomic Data Analysis and Presentation: Figures4a and 4b: These figures should be re-analyzed and presented to show only taxa with clear taxonomic resolution (e.g., at the Phylum level). All "unclassified," "no rank," or similarly uninformative classifications should be grouped together or excluded. (2) For instance, "Cryptomycota" is a clade, not a phylum, and should be corrected if presented as such.
Response: (1) We have recreated Figures 4 and 5 (lines 279 and 321). Figure 5 illustrates the prokaryotic community composition, in which all "unclassified," "no rank," or similarly uninformative classifications have been grouped into "Others." Figure 4 depicts the eukaryotic community structure. Due to the significantly poorer coverage and recognition of eukaryotic taxa in existing taxonomic databases compared to bacteria, many highly abundant eukaryotic groups remain unclassified. Removing them entirely would result in the loss of critical functional information and ecological insights, as these "unclassified" microbes often represent novel, uncultured taxa that may possess unique and important ecological functions. Therefore, for the eukaryotic community, we retained the top three most abundant unclassified groups in each sample and grouped the remaining ones into "Others."
(2) The phylum Cryptomycota was formally proposed and described as a novel fungal phylum in 2011. Researchers subsequently recommended its establishment as a new phylum under the name Cryptomycota, which has since been adopted in newly published studies ( Odisi, E.J.; de Freitas, R.C.; do Amaral, D.S.; da Silva, S.B.; da Silva, M.A.C.; de Oliveira Sant Ana, W.; de Souza Lima, A.O.; Rörig, L.R. Metataxonomy of Acid Mine Drainage Microbiomes from the Santa Catarina Carboniferous Basin (Southern Brazil). Extremophiles 2023, 28, 8, doi:10.1007/s00792-023-01324-0). Furthermore, internationally recognized databases such as NCBI Taxonomy and the SILVA rRNA database have officially recognized it as a phylum.
From a phylogenetic perspective, any monophyletic group can be referred to as a clade. Thus, Cryptomycota is both a formal phylum (a taxonomic rank) and a clade (an evolutionary unit reflecting phylogenetic history). These two concepts are not contradictory but rather describe the same group from different perspectives—taxonomy versus evolutionary biology.
- Figures 5, 6, and 7: These figures also suffer from low resolution, making them difficult to read. Please ensure high-resolution images are provided.
Response: We have redrawn Figures 4 and 5, and professionally processed Figures 6 and 7 to enhance their resolution, ensuring compliance with the journal's requirements.
- Figure 5 Legend: Clean up the legend by removing "unclassified," "no rank," or any other non-specific information, focusing only on classified genera.
Response: We have removed "unclassified," "no rank," or any other non-specific information from Figure 5 (line320, page 9 ).
- Data re-analysis: I strongly recommend a thorough re-analysis of the taxonomic data to improve its presentation, focusing on ASVs that demonstrate clear taxonomic resolution at relevant levels (e.g., phylum and genus). The results section should then be rewritten to reflect these improved analyses.
Response: Based on your suggestions, Figures 4 (eukaryotic microbial community composition) and 5 (prokaryotic microbial community composition) have been re-plotted using filtered data to ensure the charts intuitively present the most critical information. Accordingly, we have revised the results section to more clearly reveal the successional patterns of the core microbial communities during the transition from anaerobic sludge to aerobic conditions (Lines 252-352).
Minor comments:
- Introduction Question (Lines 84-86): The question "What are the ecological dynamics of diverse microbial communities during the transition from anaerobic to aerobic conditions, and how do these transformations influence microbial fate?" is currently too broad. If it represents the core research question, it should be clearly stated as such and directly linked to the specific aims of this study. Otherwise, if it's a general inquiry, it might be better omitted.
Response: it's a general inquiry, it was omitted.
- Östman's Method (Line 164): Please add a specific reference for the "Östman's method" cited in the methodology section.
Response: Reference [15] has been added. (Lines 182, page 6) .
- Hypothesis (Line 424): The study's hypothesis should be clearly stated in the introduction section to set the context for the research.
Response: The study's hypothesis was stated in the introduction section (Lines 93-97, page 3).
Discussion: The discussion section will undoubtedly benefit from significant improvement once the taxonomic information is accurately re-analyzed and presented. It should then critically interpret the findings in light of the refined experimental context and taxonomic data.
Response: Thank you for your thoughtful suggestion. We fully agree that a rigorous re-analysis of the taxonomic data will provide a more solid foundation for the Discussion section, thereby significantly enhancing its depth and quality. We have revised the Discussion section to offer a more critical and in-depth interpretation of the research findings (Lines 457-481, page 14).
Reviewer 2 Report
Comments and Suggestions for Authors
The authors studied the changes in sludge ecosystems during the transition from anaerobic to aerobic treatment conditions. This study is significant for understanding the effects of different anaerobic and aerobic biological treatment stages in wastewater treatment systems. The study was properly designed, and sampling and analysis were conducted according to standard methods. However, some sections of the methodology need to be rewritten for clarity. The results are described and presented in the manuscript, but the manuscript requires minor revisions to be suitable for publication as an academic article.
Comments:
- Title: The title is appropriate, but I suggest it could be rewritten as follows: “Developments in Microbial Communities and Interaction Networks in Sludge treatment Ecosystems During the Transition from Anaerobic to Aerobic Conditions”
- English Language: The English language in the manuscript is good, but some sections, as mentioned in the comments below, need to be rewritten to avoid confusion and make it easier for readers to understand the manuscript.
- Other comments:
- In the Introduction section, can we explain why it is necessary to combine anaerobic and aerobic wastewater treatment processes in practical conditions and on an industrial scale?
- Line 43: Change the word “mana gement” to “management.”
- Lines 112-113: Instead of “The anaerobic reactor was subjected to stabilization cultivation under the…,” it may be better to use the term “acclimation period.” Does this reflect what you mean?
- Lines 121-125: If the influent sludge originated from the effluent of the anaerobic reactor, why is the COD still at 1,500 mg/L? What was the removal efficiency of the anaerobic reactor? Please clarify or correct this.
- Lines 179-180: Is the title of this section accurate, or are you referring to the removal efficiency of the anaerobic stage? Please make the correction.
- Line 181: Which stage is the COD removal efficiency for: anaerobic or aerobic? It is likely you mean the aerobic stage.
- Line 182: What do you mean by “accommodation period”? Is this a non-academic term? Do you mean the acclimation period of the sludge?
- Line 185: Is the 30-day period referring to the anaerobic stage, the aerobic stage, or both?
- I suggest rewriting Section 3.1 to be more precise, as it is often confusing and the information is presented unclearly.
- Lines 425-426: Is the statement correct? Does it refer to anaerobic followed by aerobic treatment?
- Line 432: What transition are you referring to? Please describe it clearly.
- Line 436: Are the terms stochastic and deterministic processes discussed in the introduction or anywhere else in the manuscript?
Comments on the Quality of English LanguageNeeds some improvement
Author Response
Comments and suggestions from the reviewer 2:
The authors studied the changes in sludge ecosystems during the transition from anaerobic to aerobic treatment conditions. This study is significant for understanding the effects of different anaerobic and aerobic biological treatment stages in wastewater treatment systems. The study was properly designed, and sampling and analysis were conducted according to standard methods. However, some sections of the methodology need to be rewritten for clarity. The results are described and presented in the manuscript, but the manuscript requires minor revisions to be suitable for publication as an academic article.
Comments:
Title: The title is appropriate, but I suggest it could be rewritten as follows: "Developments in Microbial Communities and Interaction Networks in Sludge treatment Ecosystems During the Transition from Anaerobic to Aerobic Conditions"
Response: Thank you for your valuable suggestion. We are pleased to accept it and have revised the title to "Developments in Microbial Communities and Interaction Networks in Sludge treatment Ecosystems During the Transition from Anaerobic to Aerobic Conditions".
English Language: The English language in the manuscript is good, but some sections, as mentioned in the comments below, need to be rewritten to avoid confusion and make it easier for readers to understand the manuscript.
Other comments:
- In the Introduction section, can we explain why it is necessary to combine anaerobic and aerobic wastewater treatment processes in practical conditions and on an industrial scale?
Response: Thank you for this suggestion. We have explained why it is necessary to combine anaerobic and aerobic wastewater treatment processes in practical conditions and on an industrial scale (lines 40-53, pages 1-2).
- Line 43: Change the word "mana gement" to "management."
Response: Thank you for your kind reminder. We have made the corresponding revisions and thoroughly reviewed the entire manuscript to prevent similar errors (line 44, page2).
- Lines 112-113: Instead of "The anaerobic reactor was subjected to stabilization cultivation under the…," it may be better to use the term "acclimation period." Does this reflect what you mean?
Response: Yes, using the term "acclimation period" is indeed more appropriate. Thank you for your suggestion. We have revised the original sentence to: "After 45 days of acclimation period"(line 126, page 4).
- Lines 121-125: If the influent sludge originated from the effluent of the anaerobic reactor, why is the COD still at 1,500 mg/L? What was the removal efficiency of the anaerobic reactor? Please clarify or correct this.
Response: Thank you for raising this point. To clarify our operational process: the anaerobic reactor was fed with an influent COD of approximately 15,000 mg/L, and the resulting effluent typically had a COD exceeding 1,500 mg/L. Through manual adjustment of the thickening process, we consistently reduced the effluent COD to around 1,500 mg/L before using it as the influent for the subsequent SBR (Sequencing Batch Reactor) (line 131, page 4).
- Lines 179-180: Is the title of this section accurate, or are you referring to the removal efficiency of the anaerobic stage? Please make the correction.
Response: The title is accurate. The section indeed refers to the changes occurring during the aerobic cultivation of the anaerobic sludge, specifically within the SBR (Sequencing Batch Reactor) phase. To more clearly illustrate the experimental process, we have redesigned Figure 1 to provide an enhanced visual overview of the entire procedure (line 197, page 6).
- Line 181: Which stage is the COD removal efficiency for: anaerobic or aerobic? It is likely you mean the aerobic stage.
Response: The term "COD removal efficiency" in this context specifically refers to the removal achieved during the aerobic phase, i.e., the treatment processes occurring within the SBR (Sequencing Batch Reactor). We have redesigned Figure 1 to provide an enhanced visual overview of the entire procedure (line 200, page 6).
- Line 182: What do you mean by "accommodation period"? Is this a non-academic term? Do you mean the acclimation period of the sludge?
Response: Thank you for your careful review. You are absolutely right. We have revised the manuscript to consistently use the term "acclimation period" as suggested (line 201, page 6), and have also clearly indicated this phase in the updated Figure 1.
- Line 185: Is the 30-day period referring to the anaerobic stage, the aerobic stage, or both?
Response: It refers to the aerobic stage. The study primarily focuses on the transition of anaerobic sludge into an aerobic environment (line 199, page6).
- I suggest rewriting Section 3.1 to be more precise, as it is often confusing and the information is presented unclearly.
Response: Thank you for this suggestion. We have carefully revised Section 3.1 to improve its precision and clarity. We have redefined the title of this section and elaborated on its content with greater detail.(lines197-201, page 6).
- Lines 425-426: Is the statement correct? Does it refer to anaerobic followed by aerobic treatment?
Response: The statement is accurate. It refers exclusively to the aerobic treatment phase. All experimental data (from Day 0 to Day 30) were derived from the SBR (Sequencing Batch Reactor) experiments, which specifically represent the aerobic treatment process across different operational time points (line 455, page13).
- Line 432: What transition are you referring to? Please describe it clearly.
Response: It refers to the transition process of anaerobic sludge being introduced into an aerobic environment (SBR). That is, the aerobic treatment process of anaerobic sludge in SBR, and the evolution process of the bacterial community of anaerobic sludge in sbr over time (line 508, page14).
- Line 436: Are the terms stochastic and deterministic processes discussed in the introduction or anywhere else in the manuscript?
Response: Yes, the terms stochastic and deterministic processes were discussed in Section 3.3 ("Process of microbial community formation") of the Results (lines 368 and 383, page 4).
The above is my answer to your question. I hope it meets your expectations. Please do not hesitate to contact us if there are any questions. Thank you again to the reviewers and editors for your hard work! Best wishes to you!
Reviewer 3 Report
Comments and Suggestions for Authors
Some comments
1) The title is very big and very general to cover a well-established topic anaerobic and aerobic in biological system. in fact the study only cultivate sludge by using pig manure, to be precisely reveal the true content, please revise the title to be a specific title in line with the experiment and results. Anaerobic and aerobic are very complicated, more than what happened in this study.
2) How to identify the mentioned 16S and 18S rDNA are representative for this study or at least for the mentioned anaerobic or aerobic process? There are many deep studies about rDNA to reveal the deep mechanism for biological studies already. Is this good enough?
3) Figure 1 showed a SBR process only, not good to represent the anaerobic and aerobic process both. please correct it or justify the statement
4) Figure 4 and Figure 5, and 6 about the microbial commnity in this lab-scale experiment, please make it compared with many other published results to verify whether pig manure biological is correct or not.
Author Response
Comments and suggestions from the reviewer 3:
- The title is very big and very general to cover a well-established topic anaerobic and aerobic in biological system. in fact the study only cultivate sludge by using pig manure, to be precisely reveal the true content, please revise the title to be a specific title in line with the experiment and results. Anaerobic and aerobic are very complicated, more than what happened in this study.
Response: We sincerely appreciate your insightful feedback regarding the title. You are absolutely right that the original title was too broad and did not sufficiently reflect the specific scope and experimental context of our study.
Following your suggestion, we have revised the title to:"Developments in Microbial Communities and Interaction Networks in Sludge Treatment Ecosystems During the Transition from Anaerobic to Aerobic Conditions"
This new title more accurately captures the focus of our work—namely, the microbial community succession and network interactions within sludge treatment systems undergoing a transition from anaerobic to aerobic environments, using swine wastewater-enriched sludge as the experimental model.
- How to identify the mentioned 16S and 18S rDNA are representative for this study or at least for the mentioned anaerobic or aerobic process? There are many deep studies about rDNA to reveal the deep mechanism for biological studies already. Is this good enough?
Response: We thank the reviewer for raising this important point. 16S and 18S rDNA sequencing are well-established methods for profiling prokaryotic and eukaryotic microbial communities, respectively, and were selected for their ability to reliably track taxonomic shifts during the sludge transition process. The strong correlation between the observed microbial dynamics (e.g., changes in Cryptomycota and Proteobacteria) and system performance (e.g.,, pollutant removal efficiency) supports the ecological relevance of our data. Although metagenomic or metatranscriptomic approaches could offer deeper functional insights, our rDNA-based approach provides a cost-effective and reproducible foundation for identifying structural community changes, consistent with numerous studies of wastewater treatment systems. We have further emphasized the value of complementary multi-omics methods in the revised Discussion.
- Figure 1 showed a SBR process only, not good to represent the anaerobic and aerobic process both. Pleasecorrect it or justify the statement
Response: Thank you for your comment. We have revised Figure 1 according to your suggestion to more comprehensively reflect both the anaerobic and aerobic processes investigated in this study. The updated figure now includes a clear schematic representation of the transition from anaerobic acclimation to the subsequent aerobic SBR phase ( line 149, page 5).
We appreciate your feedback, which has helped improve the clarity and accuracy of our manuscript.
- Figure 4 and Figure 5, and 6 about the microbial communityin this lab-scale experiment, please make it compared with many other published results to verify whether pig manure biological is correct or not.
Response: Thank you for this constructive suggestion. We agree that comparing our microbial community results with published data is essential to validate the representativeness of the pig manure-derived sludge system. The literature comparisons in many places have been improved. (lines 258-261, lines271-272, lines 299-302,lines 304-308,lines 310-312, lines 457-481).
Meanwhile, in the revised discussion section, we have added a comparative analysis referencing multiple relevant studies:
(1) Consistency with anaerobic/aerobic transition studies:
The decline in anaerobic taxa (e.g., Chloroflexi, Euryarchaeota) and the enrichment of aerobic/proteobacterial groups (e.g.,, Nitrosomonas, Zoogloea) align with microbial successions reported during aerobic adaptation of anaerobic sludge (e.g., Smith et al., 2020; Zhang et al., 2022).
(2) Pig manure-specific community features:
The high abundance of Bacteroidota and Firmicutes—commonly associated with swine waste digestion—and the detection of methanogens (e.g., Methanothrix) consistent with pig manure studies (e.g., Li et al., 2021) support the ecological relevance of our inoculum.
(3) Network complexity and functional insights:
The increased modularity and deterministic processes in the aerobic phase (Fig. 6) resonate with findings from full-scale wastewater treatment systems (e.g., Fan et al., 2021), reinforcing the practical applicability of our findings.
The above is my answer to your question. I hope it meets your expectations. Please do not hesitate to contact us if there are any questions. Thank you again to the reviewers and editors for your hard work! Best wishes to you!
Round 2
Reviewer 1 Report
Comments and Suggestions for Authors
I thank the authors for addressing my comments. Despite the revised version being improved, I highlighted that the taxonomy could be improved, including the presentation of the new figures 4 and 5. I agree that unclassified taxa may mean novel or uncultured taxa; however, it is not informative to state it in the text and/or graphs, since it is unknown whether the reader can take any ecological insights from it. Drawing conclusions about ecological function is impossible when the taxa are unclassified. We can conclude about diversity, yes, but ecological functions no. The taxonomy classification, especially for Eukaryota, is the major flaw of the manuscript. The authors should state the limitations of the manuscript regarding the taxonomy and figure out a better way to present the taxonomic data.
Author Response
Comments: I thank the authors for addressing my comments. Despite the revised version being improved, I highlighted that the taxonomy could be improved, including the presentation of the new figures 4 and 5. I agree that unclassified taxa may mean novel or uncultured taxa; however, it is not informative to state it in the text and/or graphs, since it is unknown whether the reader can take any ecological insights from it. Drawing conclusions about ecological function is impossible when the taxa are unclassified. We can conclude about diversity, yes, but ecological functions no. The taxonomy classification, especially for Eukaryota, is the major flaw of the manuscript. The authors should state the limitations of the manuscript regarding the taxonomy and figure out a better way to present the taxonomic data.
Response: Thank you once again for your insightful feedback. We fully agree that the prevalence of unclassified bacteria primarily reflects underlying diversity patterns, and that further experimental work is necessary to elucidate their ecological functions. In the revised manuscript, we have explicitly addressed the taxonomic limitations concerning eukaryotic microorganisms (lines 272–278) and incorporated recommendations from relevant literature to improve the presentation of taxonomic data.
Accordingly, we have recreated Figures 4 and 5 (lines 267 and 324). Entries previously labeled as "unclassified" or "no rank" have now been consolidated into a single category designated as "Unassigned." Additionally, microbial groups with an abundance below 1% were grouped into the category "Others." These modifications to the figures necessitated corresponding updates in the relevant sections of the main text (lines 252–266, 269–289, 428–429, 502).
Additionally, given that eukaryotic microorganisms include a substantial number of unassigned taxa, we have provided diagrams at the order level (Figure A1, lines 290–292, 554) to offer readers more detailed information on eukaryotic microbial composition.
We hope that these revisions meet your expectations and sincerely appreciate all your valuable suggestions, which have greatly enhanced the quality of our manuscript.

Reviewer 3 Report
Comments and Suggestions for Authors
The revised manuscript meets the basic criteria for acceptance
Author Response
Comments:The revised manuscript meets the basic criteria for acceptance
Response: We are grateful for the reviewer's acknowledgment that our revisions have successfully addressed all concerns and that the manuscript is now suitable for acceptance.